# Nonlinear dynamic characterization of two-dimensional materials

D. Davidovikj [1], F. Alijani [2], S.J. Cartamil-Bueno [1], H.S.J. van der Zant [1], M. Amabili [3] &
P.G. Steeneken [1,2]

Owing to their atomic-scale thickness, the resonances of two-dimensional (2D) material membranes show signatures of nonlinearities at forces of only a few picoNewtons. Although the linear dynamics of membranes is well understood, the exact relation between the nonlinear response and the resonator's material properties has remained elusive. Here we show a method for determining the Young's modulus of suspended 2D material membranes from their nonlinear dynamic response. To demonstrate the method, we perform measurements on graphene and $MoS_2$ nanodrums electrostatically driven into the nonlinear regime at multiple driving forces. We show that a set of frequency response curves can be fitted using only the cubic spring constant as a fit parameter, which we then relate to the Young's modulus of the material using membrane theory. The presented method is fast, contactless, and provides a platform for high-frequency characterization of the mechanical properties of 2D materials.

[1] Kavli Institute of Nanoscience, Delft University of Technology, Lorentzweg 1, 2628 CJ Delft, The Netherlands. [2] Department of Precision and Microsystems Engineering, Delft University of Technology, Mekelweg 2, 2628 CD Delft, The Netherlands. [3] Department of Mechanical Engineering, McGill University, 817 Sherbrooke Street West, Montreal, QC, Canada H3A 2K6. Correspondence and requests for materials should be addressed to D.D. (email: d.davidovikj@tudelft.nl)

The remarkable mechanical properties of two-dimensional (2D) material membranes have sparked interest for potential uses as pressure[1, 2], gas[3, 4], and mass[5, 6] sensors. For such applications, it is essential to have accurate methods for determining their mechanical properties. One of the most striking properties of these layered materials is their high Young's modulus. To measure the Young's modulus, a number of static deflection techniques have been used, including atomic force microscopy (AFM)[7–10], the pressurized blister test[11], and the electrostatic deflection method[12, 13]. The most widely used method is AFM, where by performing a nanoindentation measurement at the center of a suspended membrane, its pre-tension ($n_0$) and Young's modulus ($E$) are extracted from the force–deflection curve. Despite the large number of experimental and theoretical studies[14, 15], the exact physics behind the elasticity of 2D materials is still a subject of debate[16]. This debate is mainly motivated by the large spread in values reported in literature ($E_{graphene} = 430–1200\,GPa$)[14], which has been attributed to variations in the material properties and fabrication techniques[17]. As a consequence, there is a significant interest in methods for characterizing the mechanical properties of 2D materials.

Although AFM has been the method of choice for static studies, laser interferometry has proven to be an accurate tool for the dynamic characterization of suspended 2D materials, with dynamic displacement resolutions better than $20\,fm/\sqrt{Hz}$ at room temperature[18–20]. As for very thin structures the resonance frequency is directly linked to the pre-tension in the membrane, these measurements have been used to mechanically characterize 2D materials in the linear limit[18, 19, 21, 22]. At high vibrational amplitudes, nonlinear effects start playing a role, which have lately attracted a lot of interest[23–28]. In particular, Duffing-type nonlinear responses have been regularly observed[18–20, 29, 30]. These geometrical nonlinearities, however, have never been related to the intrinsic material properties of the 2D membranes.

Here, we introduce a method for determining the Young's modulus of 2D materials by fitting their forced nonlinear Duffing response. Using nonlinear membrane theory, we derive an expression that allows us to relate the fit parameters to both the pre-tension and Young's modulus of the material. The proposed method offers several advantages. First, the excitation force is purely electrostatic, requiring no physical contact with the membrane that can influence its shape[31, 32]. Second, the on-resonance dynamic operation significantly reduces the required actuation force, compared with the static deflection methods. Third, the high-frequency resonance measurements allow for fast testing by averaging over millions of deflection cycles per second, using mechanical frequencies in the MHz range. Lastly, the membrane motion is so fast that slow viscoelastic deformations due to delamination, slippage, and wall adhesion effects are strongly reduced. To demonstrate the method, we measure and analyze the nonlinear dynamic response of suspended 2D nanodrums.

## Results

**Measurements**. The samples consist of cavities on top of which exfoliated flakes of 2D materials are transferred using a dry transfer technique[33]. One of the measured devices, a few-layer (FL) graphene nanodrum, is shown in the inset of Fig. 1a. The measurements are performed in vacuum at room temperature. Electrostatic force is used to actuate the membrane and a laser interferometer is used to detect its motion, as described in refs. [18–20, 22]. A schematic of the measurement setup

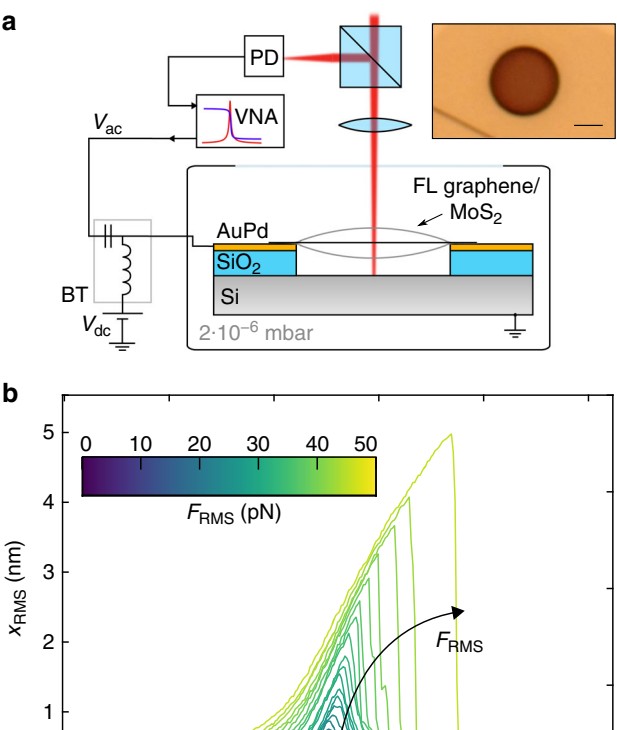

**Fig. 1** Measurement setup and measured frequency response of device 1. **a** Schematic of the measurement setup: a laser interferometer setup is used to read out the motion of the nanodrum. The Si substrate is grounded and, using a bias-tee (BT), a combination of ac voltage and dc voltage is applied to electrostatically actuate the motion of the drum. This motion modulates the reflected laser intensity and the modulation is read out by a photodiode. Inset: an optical image of a FL graphene nanodrum (scale bar: 2 µm). **b** Frequency response curves of the calibrated root-mean-square (RMS) motion amplitude for increasing electrostatic driving force. The onset of nonlinearity is visible above $F_{RMS} = 15\,pN$. The color of the curves indicates the corresponding driving force

is shown in Fig. 1a. The details on the sample preparation and measurement setup are described in the "Methods" section below.

Figure 1b shows a set of calibrated frequency response curves of the fundamental mode of a graphene drum (device 1, with thickness $h = 5\,nm$ and radius $R = 2.5\,µm$) driven at different ac voltages ($V_{ac}$). The dc voltage is kept constant ($V_{dc} = 3\,V$) throughout the entire measurement with $V_{dc} \gg V_{ac}$. All measurements are taken using upward frequency sweeps. The RMS force $F_{RMS}$ is the root-mean-square (RMS) of the electrostatic driving force. For high-driving amplitudes ($F_{RMS} > 15\,pN$), the resonance peak starts to show a nonlinear hardening behavior, which contains information on the cubic spring constant of the membrane.

**Fitting the nonlinear response**. We can approximate the nonlinear response of the fundamental resonance mode by the Duffing equation (Supplementary Note 1):

$$m_{eff}\ddot{x} + c\dot{x} + k_1 x + k_3 x^3 = \xi F_{el}\cos(\omega t), \qquad (1)$$

where $x$ is the deflection of the membrane's center, $c$ is the

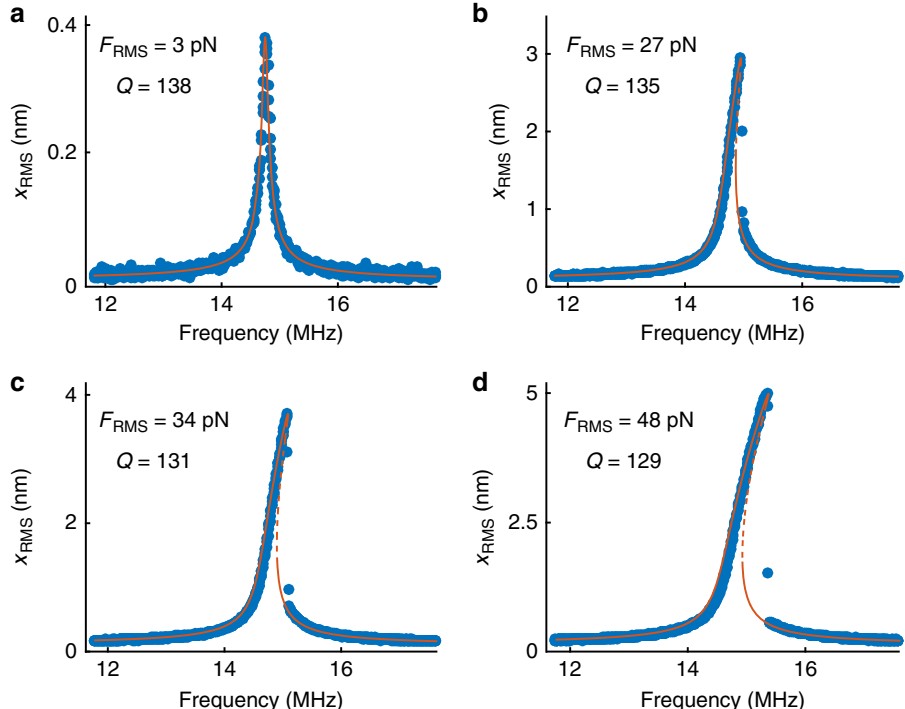

**Fig. 2** Measurements and fits of the nonlinear frequency response curves of device 1. Measured traces (blue scatter plot) and the corresponding fits (red curves) showing both the stable (solid line) and the unstable (dashed line) solutions of the Duffing equation. **a–d** Frequency response curves of the device from Fig. 1 at four different driving forces, denoted in the top left corner of each panel, along with the extracted $Q$-factors. The extracted cubic spring constant is $k_3 = 1.35 \times 10^{15}$ N m$^{-3}$

damping constant, $k_1$ and $k_3$ are the linear and cubic spring constants, and $m_{eff} = \alpha m$ and $\xi F_{el}$ are the mass and the applied electrostatic force corrected by factors ($\alpha$ and $\xi$) that account for the mode shape of the resonance (for a rigid-body vertical motion of the membrane $\alpha$ and $\xi$ are both 1). As shown in the Supplementary Note 1, for the fundamental mode of a fixed circular membrane $\xi = 0.432$ and $\alpha = 0.269$. The parameters in the Duffing equation (Eq. (1)) are related to the resonance frequency $\omega_0$ ($\omega_0 = 2\pi f_0$) and the $Q$-factor by $Q = \omega_0 m_{eff}/c$ and $\omega_0^2 = k_1/m_{eff}$.

The fundamental resonance frequency ($f_0 = 14.7$ MHz) is extracted from the linear response curves at low-driving powers (Fig. 1b), and is directly related to the pre-tension ($n_0$) of the membrane: $n_0 = 0.69\pi^2 f_0^2 R^2 \rho h$, where $\rho$ is the mass density of the membrane (for device 1, $n_0 = 0.107$ N m$^{-1}$). To fit the set of nonlinear response curves, the steady-state solution of the Duffing equation (Eq. (1)) is converted to a set of algebraic equations using the harmonic balance method (Supplementary Note 2). Using these equations, the entire set of curves can then be fitted by a least-squares optimization algorithm. As $N$ curves are fitted simultaneously, the expected fitting error is roughly a factor $\sqrt{N}$ lower than that of single curve fit.

The $Q$-factor is implicitly related to $k_3$ by a function $Q_i = Q_i(k_3, A_{max,i}, F_{el,i})$, where $A_{max,i}$ are the peak amplitudes and $F_{el,i}$ are the driving force amplitudes for each of the measured curves[34, 35] (Supplementary Note 2). The amplitudes $A_{max,i}$ are found from the experimental data and the whole dataset is fitted using a single fit parameter: the cubic spring constant $k_3$. The results of this procedure are presented in Fig. 2a–d, which shows four frequency response curves and their corresponding fits. The solutions of the steady-state amplitude for the Duffing equation (red curves in Fig. 2) are plotted by finding the positive real roots

$x^2$ of:

$$\xi^2 F_{el}^2 = \left( \omega^2 c^2 + m_{eff}^2 \left( \omega^2 - \omega_0^2 \right)^2 \right) x^2$$

$$-\frac{3}{2} m_{eff} \left( \omega^2 - \omega_0^2 \right) k_3 x^4 + \frac{9}{16} k_3^2 x^6. \qquad (2)$$

A good agreement between fits and data is found using the single extracted value $k_3 = 1.35 \times 10^{15}$ N m$^{-3}$, which demonstrates the correspondence between the measurement and the underlying physics. We note that at higher driving amplitudes, we also observe a reduction in the $Q$-factor (by nearly 10% at the highest measured driving amplitude). This can be a signature of nonlinear damping mechanisms, which is in line with previously reported measurements on graphene mechanical resonators[23, 24, 36]. In the following section, we will lay out the theoretical framework to relate the extracted cubic spring constant $k_3$ to the Young's modulus of the membrane.

**Theory.** The nonlinear mechanics of a membrane can be related to its material parameters via its potential energy. The potential energy of a radially deformed circular membrane with isotropic material properties can be approximated by a function of the form:

$$U = \frac{1}{2} C_1 n_0 x^2 + \frac{1}{4} C_3(\nu) \frac{Eh\pi}{R^2} x^4, \qquad (3)$$

where $R$ and $h$ are the membrane's radius and thickness, respectively. Bending rigidity is neglected, which is a good approximation for $h/R < 0.001$[37]. $C_1$ and $C_3(\nu)$ are dimensionless functions that depend on the deformed shape of the membrane

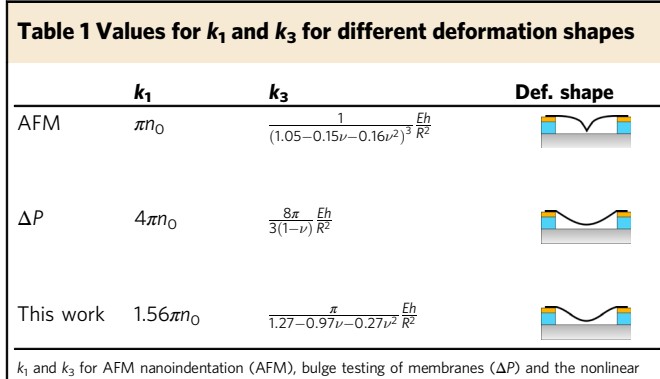

**Table 1 Values for $k_1$ and $k_3$ for different deformation shapes**

| | $k_1$ | $k_3$ | Def. shape |
|---|---|---|---|
| AFM | $\pi n_0$ | $\frac{1}{(1.05-0.15\nu-0.16\nu^2)^3}\frac{Eh}{R^2}$ | |
| $\Delta P$ | $4\pi n_0$ | $\frac{8\pi}{3(1-\nu)}\frac{Eh}{R^2}$ | |
| This work | $1.56\pi n_0$ | $\frac{\pi}{1.27-0.97\nu-0.27\nu^2}\frac{Eh}{R^2}$ | |

$k_1$ and $k_3$ for AFM nanoindentation (AFM), bulge testing of membranes ($\Delta P$) and the nonlinear dynamics method (this work) for the fundamental resonance mode. The corresponding deformation shape, which determines the functional dependence of $k_1$ and $k_3$, is shown on the right

and the Poisson's ratio $\nu$ of the material. The term in Eq. (3) involving $C_1$ represents the energy required to stretch a membrane under a constant tensile pre-stress, the $C_3$ term signifies that the tension itself starts to increase for large membrane deformations. The out-of-plane mode shape for the fundamental resonance mode of a circular membrane is described by a zero-order Bessel function of the first kind ($J_0(r)$). Numerical calculations of the potential energy (3) of this mode give $C_1 = 1.56\pi$ and $C_3(\nu) = 1/(1.269 - 0.967\nu - 0.269\nu^2)$ (Supplementary Note 1; Supplementary Fig. 1). Using Eq. (3), the nonlinear force–deflection relation of circular membranes is given by

$$F = \frac{dU}{dx} = k_1 x + k_3 x^3 = C_1 n_0 x + C_3(\nu)\frac{Eh\pi}{R^2}x^3. \quad (4)$$

The functions $C_1$ and $C_3$ have previously been determined for the potential energies of statically deformed membranes by AFM[9, 38] and uniform gas pressure[39, 40]. Their functional dependence depends entirely on the shape of the deformation of the membrane. In Table 1, we summarize the functional dependences of $k_1$ and $k_3$ for the three types of membrane deformation.

By combining Eq. (4) with the obtained functions for $C_1$ and $C_3$ from Table 1 (last row), the Young's modulus $E$ can be determined from the cubic spring constant $k_3$ by

$$E = \frac{(1.27 - 0.97\nu - 0.27\nu^2)R^2}{\pi h}k_3. \quad (5)$$

From this equation, with the value of $k_3$ extracted from the fits, a Young's modulus of $E = 594 \pm 45$ GPa is found, which is in accordance with literature values, which range from 430 to 1200 GPa[14, 17]. By calculating the standard deviation out of nine repeated measurements, the measurement error was determined to be 8%, which is comparable to other methods for determining the Young's modulus of 2D materials[7]. The numerical error in the Young's modulus from the fitting procedure is typically <0.5% (defined as the 95% confidence interval of the fit), as determined from the raw data and the fits, like those shown in the Supplementary Figs. 7–15. Our measurement error is therefore mainly experimental. Using $E = 594$ GPa, the nonlinear dynamic response of the system can be modeled for different driving powers and frequencies. Figure 3 shows color plots representing the RMS amplitude of the motion of the membrane center as a function of frequency and driving force. Excellent agreement is found between the experiment (Fig. 3a) and the model (Fig. 3b).

To confirm the validity of the method, we performed an AFM nanoindentation measurement on the same graphene drum. A

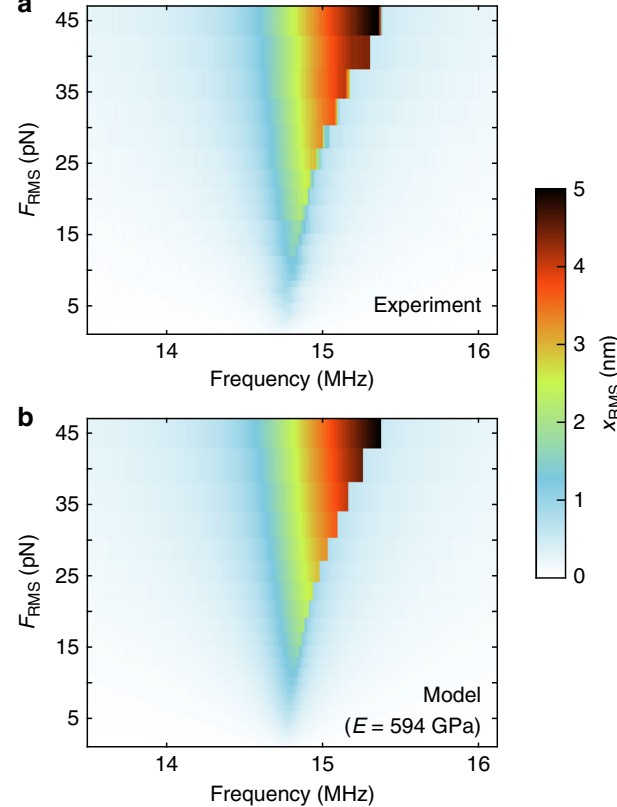

**Fig. 3** Comparison between the measured and the modeled response. Comparison of the RMS motion amplitude ($x_{RMS}$) between experiment (**a**) and model (**b**) using the fitted value for the Young's modulus ($E = 594$ GPa) for the device shown in Fig. 1 (device 1)

force–deflection measurement, taken at the center of the drum, is plotted in Fig. 4 (black dots). The curve is fitted by the AFM force–deflection equation given in Table 1, yielding $E = 591$ GPa and $n_0 = 0.093$ N m$^{-1}$ (red curve in Fig. 4). The blue curve shows the expected force–deflection curve based on the values for the Young's modulus and pre-tension extracted from the nonlinear dynamic response fits. The two curves are in close agreement.

To demonstrate the versatility of the method, additional measurements on two MoS$_2$ nanodrums from the same flake are presented in Fig. 5a and b. The extracted Young's moduli are: (a) $E = 315 \pm 23$ GPa and (b) $E = 300 \pm 18$ GPa. As with device 1, the measurement error was determined by taking the standard deviation from nine repeated measurements. These numbers are also in agreement with literature values ($E_{MoS_2} = 140 - 430$ GPa[9, 14]). The extracted pre-tension of the drums is (a) $n_0 = 0.22$ N m$^{-1}$ and (b) $n_0 = 0.21$ N m$^{-1}$.

## Discussion

There are several considerations that one needs to be aware of when applying the proposed method. In an optical detection scheme, as the one presented in this work, the cavity depth has to be optimized so that the photodiode voltage is still linearly related to the motion at high amplitudes and the power of the readout laser has to be kept low to avoid significant effects of optothermal back-action[41]. The proposed mathematical model assumes that the bending energy is much smaller than the membrane energy. This is valid for membranes under tension (thickness-to-radius ratio $h/R < 0.001$)[37], as is most often the case with suspended 2D materials[18, 19, 21]. It is noted that the electrostatic force also has a nonlinear spring-softening component due to its displacement

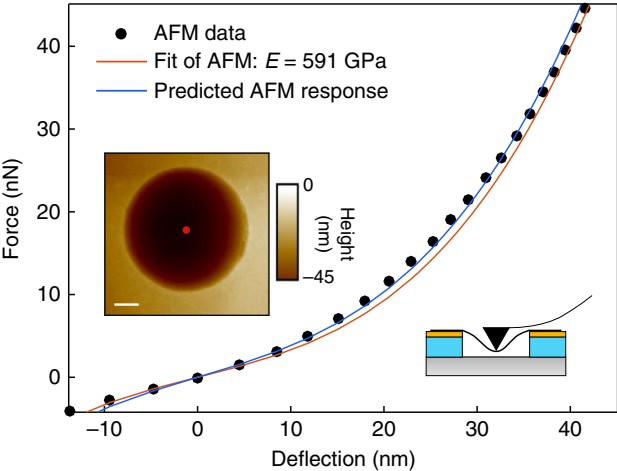

**Fig. 4** AFM force–deflection measurement. AFM force–deflection curve during tip retraction and the corresponding fit (red curve). Inset shows the AFM image of the drum (scale bar is 1 μm). The curve is taken at the center of the drum for device 1 (marked by the red dot in inset). The blue curve represents the predicted AFM response using the $n_0 = 0.107$ N m$^{-1}$ and $E = 594$ GPa, obtained from the fit of the nonlinear dynamic response

amplitude dependence. However, in the current study, the vibration amplitudes are much smaller than the cavity depth and this contribution can be safely neglected (see Supplementary Note 3 for derivation). In addition, the method requires knowledge of the mass of the resonator, which might be affected by contamination. In the presented data, the close agreement between the extracted Young's modulus and the one determined by AFM (which is independent of the mass of the membrane) suggests that the mass estimate is accurate and effects of contamination are small.

Compared to conventional mechanical characterization methods[7–13], the presented method provides several advantages. First, no physical contact to the flake is required. This prevents effects such as adhesion and condensation of liquids between an AFM tip and the membrane, that can influence the measurements. Moreover, the risk of damaging the membrane is significantly reduced. The on-resonance operation allows the usage of very small actuation forces, as the motion amplitude at resonance is enhanced by the Q-factor. Unlike AFM, where the force is concentrated in one point, here the force is more equally distributed across the membrane, resulting in a more uniform stress distribution. In addition, for resonators with a high-quality factor, the mode shape of vibrations is practically independent of the shape or geometry of the actuator.

The high-frequency nature of the presented technique is advantageous, since it allows for fast characterization of samples, and might even be extended for fast wafer-scale characterization of devices. Every point of the frequency response curve corresponds to many averages of the full force–deflection curve (positive and negative part), which reduces the error of the measurement and eliminates the need for offset calibration of the zero point of displacement[34]. The close agreement between the AFM and nonlinear dynamics value for the Young's modulus E indicates that viscoelasticity, and other time-dependent effects like slippage and relaxation, are small in graphene. Therefore, the dynamic stiffness is practically coinciding with the static stiffness. For future studies, it is of interest to apply the method to study viscoelastic effects in 2D materials, where larger differences between AFM and resonant characterization measurements are expected.

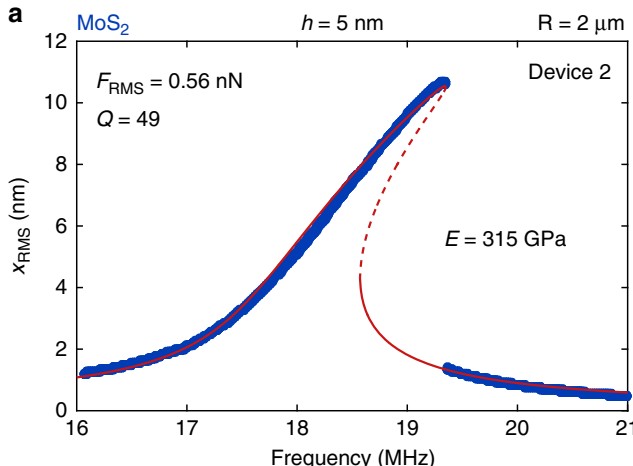

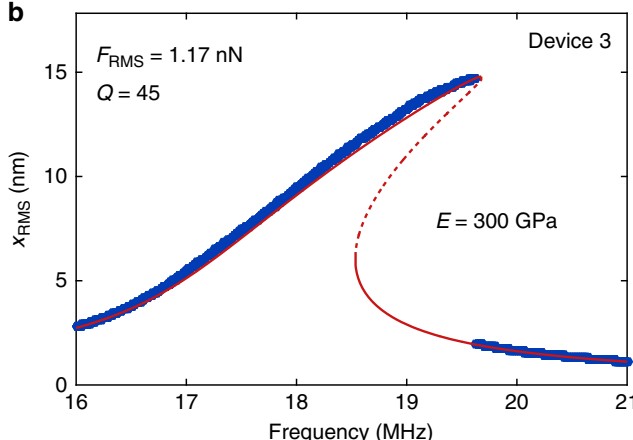

**Fig. 5** Measurements on two MoS$_2$ drums. Measurements (blue dots) and fits (drawn red curves: stable solutions; dashed red curves: unstable solutions) of two 5 nm-thick-MoS$_2$ drums with Young's moduli of: **a** device 2: 315 GPa and **b** device 3: 300 GPa

To test the robustness of the method, we perform a set of nine measurements on another graphene drum (device 4, with thickness $h = 8$ nm and radius $R = 2$ μm), under different conditions. Each of the measurements (blue and red dots in Fig. 6a) represents a fit of eight nonlinear response curves at different driving powers for a fixed dc voltage (raw data and fits are shown in Supplementary Figs. 7–15). The same set of measurements are presented in the histogram given in Fig. 6b. The extracted average value of the Young's modulus is $559 \pm 23$ GPa, which is in the same order of magnitude as the one for device 1 (where all nine measurements were taken at a single dc voltage). In Fig. 6c, we plot the raw data (black dots) and the fit (red curve) of the nonlinear response curve at $V_{dc} = 0.9$ V and $F_{RMS} = 0.42$ nN using the extracted average Young's modulus from Fig. 6a and b. The data and the fit show good agreement, which confirms that the method is robust against measurement parameter variations. The robustness of the method can also be assessed qualitatively from the effect of the Young's modulus on the linear response curve as shown in Supplementary Note 4 and Supplementary Fig. 2. There we plot the predicted response of the drum using different values of the Young's modulus to visualize its effect on the shape of the nonlinear frequency response curves.

In Table 2, we show a summary of the measurements of the four devices presented in this work. In all four cases, the error in the Young's modulus ($\sigma_E/E$) is <8% and the values of the Young's

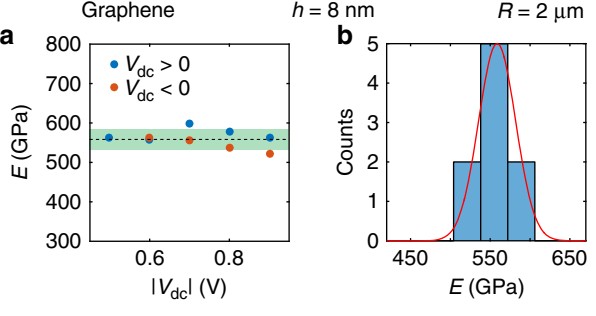

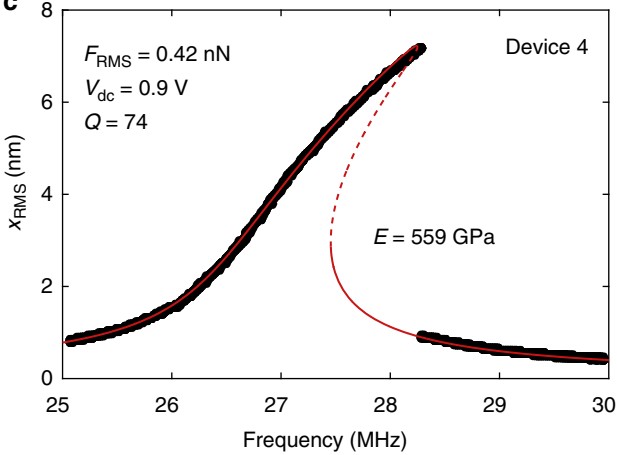

**Table 2 Summary of extracted Young's moduli and corresponding errors ($\sigma_E$) of the different samples measured in this work**

| | Material | E (GPa) | $\sigma_E$ (GPa) |
|---|---|---|---|
| Device 1 | Graphene | 594 | 45 |
| Device 2 | $MoS_2$ | 315 | 23 |
| Device 3 | $MoS_2$ | 300 | 18 |
| Device 4 | Graphene | 559 | 23 |

**Fig. 6** Analysis of the robustness and repeatability of the method using a second graphene drum (device 4). **a** Extracted Young's modulus as a function of applied dc voltage (blue dots: positive $V_{dc}$, red dots: negative $V_{dc}$). The dashed line represents the average value of the Young's modulus and the green area represents the error in the measurement ($2\sigma_E$). Each dot represents an extracted value from fitting eight nonlinear response curves at different driving powers. **b** A histogram of the values extracted from **a**. The red line is a Gaussian fit to the data. **c** Measurement (black dots) and fit (drawn red curves: stable solutions; dashed red curves: unstable solutions) of the nonlinear dynamic response of device 4, using the average value of the Young's modulus from **a**, **b**: 559 GPa

moduli of the two graphene and the two $MoS_2$ drums are within each other's error bars. The presented method can prove to be useful for fast statistical analysis of the spread in material properties[13, 42–44] and variability of device properties in future 2D material-based products.

In conclusion, we provide a contactless method for characterizing the mechanical properties of suspended 2D materials using their nonlinear dynamic response. A set of nonlinear response curves is fitted using only one fit parameter: the cubic spring constant. Mathematical analysis of the membrane mechanics is used to relate the Duffing response of the membrane to its material and geometrical properties. These equations are used to extract the pre-tension and Young's modulus of both graphene and $MoS_2$, which are in close agreement with nanoindentation experiments. The non-contact, on-resonant, high-frequency nature of the method provides numerous advantages, and makes it a powerful alternative to AFM for characterizing the mechanical properties of 2D materials. We envision applications in metrology tools for fast and non-contact characterization of 2D membranes in commercial sensors and actuators.

## Methods

**Sample fabrication**. A chip with cavities is fabricated from a thermally oxidized Si wafer, with a $SiO_2$ thickness of 285 nm, using standard lithographic and metal deposition techniques. Circular cavities are etched into the oxide by using a 100 nm gold-palladium ($Au_{0.6}Pd_{0.4}$) hard mask, which also functions as an electrical contact to the 2D flake. The final depth of the cavities is g = 385 nm and their radii are R = 2–2.5 μm. The flakes of graphene and $MoS_2$ are exfoliated from natural crystals.

**Measurement setup**. The sample is mounted in a vacuum chamber ($2 \times 10^{-6}$ mbar) to minimize damping by the surrounding gas. Using the silicon wafer as a backgate, the membrane is driven by electrostatic force and its dynamic motion is detected using a laser interferometer (see ref. [20]). The detection is performed at the center of the drum (using a laser power of 0.42 mW), using a Vector Network Analyzer (VNA). A dc voltage ($V_{dc}$) is superimposed on the ac output of the VNA ($V_{ac}$) through a bias-tee (BT), such that the small-amplitude driving force at frequency $\omega$ is given by $F_{el}(t) = \xi \varepsilon_0 R^2 \pi V_{dc} V_{ac} \cos(\omega t)/g^2$. Even though the determination of the force is mathematically straightforward, the calculated force does not always match the force felt by the resonator, because of uncertainties in determining the gap size g (due to membrane slack), the dc voltage $V_{dc}$ (due to residual charge on the 2D flake), and the capacitance of the device (due to fringe fields). To cross-check the value of the driving force, we employ a second method to determine that it is based on the peak RMS amplitude, ($x_{RMS}|_{\omega=\omega_0}$) of the calibrated linear frequency response curves using $F_{RMS} = \frac{\omega_0^2 m_{eff}}{Q} x_{RMS}$. The procedure is discussed in more detail in the Supplementary Note 5 and an example of the force derivation is shown in Supplementary Fig. 3. The measured VNA signal (in V/V) is converted to $x_{RMS}$, using a calibration measurement of the thermal motion taken with a spectrum analyzer[18, 20, 45]. The calibration procedure and the uncertainties stemming from the assumption of linear transduction are discussed in detail in Supplementary Notes 6 and 7 and Supplementary Figs. 4 and 5. The temperature increase due to laser heating is estimated in Supplementary Fig. 6.

**Data availability**. The raw data that support the findings of this study are available from the corresponding authors on request.

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

# ARTICLE

11. Koenig, S. P., Boddeti, N. G., Dunn, M. L. & Bunch, J. S. Ultrastrong adhesion of graphene membranes. *Nat. Nanotechnol.* **6**, 543–546 (2011).

12. Wong, C., Annamalai, M., Wang, Z. & Palaniapan, M. Characterization of nanomechanical graphene drum structures. *J. Micromech. Microeng.* **20**, 115029 (2010).

13. Nicholl, R. J. et al. The effect of intrinsic crumpling on the mechanics of free-standing graphene. *Nat. Commun.* **6**, 8789 (2015).

14. Castellanos-Gomez, A., Singh, V., van der Zant, H. S. J. & Steele, G. A. Mechanics of freely-suspended ultrathin layered materials. *Annalen der Phys.* **527**, 27–44 (2015).

15. Akinwande, D. et al. A review on mechanics and mechanical properties of 2D materials - graphene and beyond. *Extreme Mech. Lett.* **13**, 42–77 (2017).

16. Los, J., Fasolino, A. & Katsnelson, M. Mechanics of thermally fluctuating membranes. *npj 2D Mater. Appl.* **1**, 9 (2017).

17. Isacsson, A. et al. Scaling properties of polycrystalline graphene: a review. *2D Mater.* **4**, 012002 (2017).

18. Bunch, J. S. et al. Electromechanical resonators from graphene sheets. *Science* **315**, 490–493 (2007).

19. Castellanos-Gomez, A. et al. Single-layer $MoS_2$ mechanical resonators. *Adv. Mater.* **25**, 6719–6723 (2013).

20. Davidovikj, D. et al. Visualizing the motion of graphene nanodrums. *Nano Lett.* **16**, 2768–2773 (2016).

21. Cartamil-Bueno, S. J. et al. High-quality-factor tantalum oxide nanomechanical resonators by laser oxidation of $TaSe_2$. *Nano Res.* **8**, 2842–2849 (2015).

22. Wang, Z. et al. Black phosphorus nanoelectromechanical resonators vibrating at very high frequencies. *Nanoscale* **7**, 877–884 (2015).

23. Eichler, A. et al. Nonlinear damping in mechanical resonators made from carbon nanotubes and graphene. *Nat. Nanotechnol.* **6**, 339–342 (2011).

24. Croy, A., Midtvedt, D., Isacsson, A. & Kinaret, J. M. Nonlinear damping in graphene resonators. *Phys. Rev. B* **86**, 235435 (2012).

25. Eriksson, A., Midtvedt, D., Croy, A. & Isacsson, A. Frequency tuning, nonlinearities and mode coupling in circular mechanical graphene resonators. *Nanotechnology* **24**, 395702 (2013).

26. De Alba, R. et al. Tunable phonon-cavity coupling in graphene membranes. *Nat. Nanotechnol.* **11**, 741–746 (2016).

27. Mathew, J. P., Patel, R. N., Borah, A., Vijay, R. & Deshmukh, M. M. Dynamical strong coupling and parametric amplification of mechanical modes of graphene drums. *Nat. Nanotechnol.* **11**, 747–751 (2016).

28. Houri, S. et al. Direct and parametric synchronization of a graphene self-oscillator. *Appl. Phys. Lett.* **110**, 073103 (2017).

29. Chen, C. et al. Performance of monolayer graphene nanomechanical resonators with electrical readout. *Nat. Nanotechnol.* **4**, 861–867 (2009).

30. Chen, C. et al. Graphene mechanical oscillators with tunable frequency. *Nat. Nanotechnol.* **8**, 923–927 (2013).

31. Han, J., Pugno, N. M. & Ryu, S. Nanoindentation cannot accurately predict the tensile strength of graphene or other 2D materials. *Nanoscale* **7**, 15672–15679 (2015).

32. Vella, D. & Davidovitch, B. Indentation metrology of clamped, ultra-thin elastic sheets. *Soft Matter* **13**, 2264–2278 (2017).

33. Castellanos-Gomez, A. et al. Deterministic transfer of two-dimensional materials by all-dry viscoelastic stamping. *2D Mater.* **1**, 011002 (2014).

34. Lifshitz, R. & Cross, M. Nonlinear dynamics of nanomechanical and micromechanical resonators. *Rev. Nonlinear Dyn. Complex.* **1**, 1–52 (2008).

35. Amabili, M., Alijani, F. & Delannoy, J. Damping for large-amplitude vibrations of plates and curved panels, part 2: identification and comparisons. *Int. J. Non-Linear Mech.* **85**, 226–240 (2016).

36. Singh, V., Shevchuk, O., Blanter, Y. M. & Steele, G. A. Negative nonlinear damping of a multilayer graphene mechanical resonator. *Phys. Rev. B* **93**, 245407 (2016).

37. Mansfield, E. H. *The Bending and Stretching of Plates* (Cambridge University Press, Cambridge, 2005).

38. Komaragiri, U., Begley, M. & Simmonds, J. The mechanical response of freestanding circular elastic films under point and pressure loads. *Trans. ASME-E J. Appl. Mech.* **72**, 203–212 (2005).

39. Hencky, H. Uber den spannungszustand in kreisrunden platten mit verschwindender biegungssteiflgkeit. *Zeitschrift fur Mathematik und Physik* **63**, 311–317 (1915).

40. Boddeti, N. G. et al. Mechanics of adhered, pressurized graphene blisters. *J. Appl. Mech.* **80**, 040909 (2013).

41. Barton, R. A. et al. Photothermal self-oscillation and laser cooling of graphene optomechanical systems. *Nano Lett.* **12**, 4681–4686 (2012).

42. López-Polín, G. et al. Increasing the elastic modulus of graphene by controlled defect creation. *Nat. Phys.* **11**, 26–31 (2015).

43. Gornyi, I., Kachorovskii, V. Y. & Mirlin, A. Anomalous hooke's law in disordered graphene. *2D Mater.* **4**, 011003 (2016).

44. Nicholl, R. J. T., Lavrik, N. V., Vlassiouk, I., Srijanto, B. R. & Bolotin, K. I. Hidden area and mechanical nonlinearities in freestanding graphene. *Phys. Rev. Lett.* **118**, 266101 (2017).

45. Hauer, B., Doolin, C., Beach, K. & Davis, J. A general procedure for thermomechanical calibration of nano/micro-mechanical resonators. *Ann. Phys.* **339**, 181–207 (2013).

## Acknowledgements

This work was supported by the Netherlands Organisation for Scientific Research (NWO/OCW), as part of the Frontiers of Nanoscience (NanoFront) program and the European Union Seventh Framework Programme under grant agreement no. 604391 Graphene Flagship.

## Author contributions

D.D., F.A. and P.G.S. conceived the experiment; D.D. fabricated the samples and conducted the measurements; S.J.C.-B. performed the AFM nanoindentation measurements; F.A. built the nonlinear mechanics model; M.A. developed the identification algorithm; F.A. and D.D. performed the fitting; data analysis and interpretation were done by D.D., F.A., H.S.J.v.d.Z., M.A. and P.G.S.; H.S.J.v.dZ. and P.G.S. supervised the project; the manuscript was written by D.D. and P.G.S. with inputs from all authors.

## Additional information

**Competing interests:** The authors declare no competing financial interests.

