## [Peer Review File · Nature Communications]

Reviewers' comments:

Reviewer #1 (Remarks to the Author):

A good paper showcasing the nonlinearity in resonators made from 2-D devices. Optical measurements have been used, which are always hard to do; however, would help the field to grow. There are two principal concerns. One is the repeatability of the process and variability of the devices measured. One would expect that by measuring from several devices, the k_1 and k_3 would show a Gaussian distribution; however, the extracted Young's modulus will be within a much finer error. It would be nice to see what is the sensitivity and resolution of this measurement setup. Hence, I would highly recommend testing it with large number of devices. Second, it is a fixed-fixed beam. Why can we not use its linear region of operation with small excitation to find Young's modulus as is the case with most other resonators. The nonlinearity adds interesting aspects and is worth studying; however, its variability and difficulties of stability make it even more difficult to apply in real practice.

Reviewer #2 (Remarks to the Author):

This is an important paper that describes the use of resonance excitation and laser interferometry to evaluate the Young's modulus of 2D materials. The nonlinear response of the material and a single-parameter fit to the Duffing equation are used. As the authors point out, even though the AFM is most often used, there are outstanding issues with the use of the AFM to extract the Young's modulus of 2D materials. The method proposed in this paper provides an attractive alternative. I suggest the paper be accepted for publication after the authors have addressed the following.

In the Theory section the authors say bending rigidity is neglected, which is a good approximation for $h/R > 1/1000$. However, in the Discussion, they say this approximation is valid for membranes under tension (thickness-to-radius ratio $h/R < 0.001$). The latter is correct.

Reviewer #3 (Remarks to the Author):

The authors report on a new method to determine the Young's modulus of thin membranes based on graphene and MoS₂. The method consists in measuring the nonlinear dynamic response of the resonator. However, it is not clear whether this method is more reliable than previous methods, since the Young's modulus obtained with their new method is similar to that obtained with the previously existing AFM method (Figs. 2 and 4). Moreover, the authors write in the introduction: "values [of the Young's modulus] reported in literature show large spreads [14], which may result [...] from imperfections of the measurement method itself"; but the authors fail to show measurements on multiple graphene and MoS₂ resonators in order to show that they obtain the same Young's modulus. In addition, the authors write in the introduction "One of the most striking properties of these ultra-thin materials is their high Young's modulus", suggesting that the high Young modulus of thin materials is related to their atomic scale thickness. But this is not correct, since the Young modulus of graphene is about the same as that of graphite. Finally, this method may be interesting as a rapid characterization of the Young's modulus, but this is something interesting for specialists only. For these reasons, I recommend rejection.

Here are additional comments:

1- The authors start the abstract with "Due to their atomic-scale thickness, the resonances of 2D

material membranes show signatures of nonlinearities at amplitudes of only a few nanometers.” This is a strange sentence since thicker resonators also feature nonlinear resonances at amplitudes of a few nanometers.

2- How reliable is their method in quantifying the Young’s modulus? It seems that the value of the Young’s modulus depends on the effective mass and the electrostatic force. The authors do not include the mass of the contamination adsorbed on the surface, which is known to considerably contribute to the mass. It is not clear whether a precise determination of the electrostatic force can be made.

3- The authors should include a detailed description of the calibration of the displacement, which is probably not straightforward to do. Again, this will contribute to the uncertainty of the estimation of the Young’s modulus.

4- How robust is the determination of the Young’s modulus considering that the Q-factor gets reduced at large drives?

Reviewers' comments and our response in blue:

Reviewer #1 (Remarks to the Author):

A good paper showcasing the nonlinearity in resonators made from 2-D devices. Optical measurements have been used, which are always hard to do; however, would help the field to grow. There are two principal concerns. One is the repeatability of the process and variability of the devices measured. One would expect that by measuring from several devices, the k_1 and k_3 would show a Gaussian distribution; however, the extracted Young's modulus will be within a much finer error. It would be nice to see what is the sensitivity and resolution of this measurement setup. Hence, I would highly recommend testing it with large number of devices.

We thank the reviewer for his/her positive comments.

We agree with the reviewer that it is very important to determine the repeatability and error of the measurement process. For a pure characterisation of the repeatability and measurement error, we need to exclude sample-to-sample variations and characterise the same drum over a set of measurements. This was already done for the two presented samples, where the error was determined as the standard deviation of a set of nine measurements under the same conditions. We have now made this more clear in the text.

To test the robustness of our method, we also include a statistical analysis of one of the new samples under different measurement conditions (different V_{dc}). We show statistics based on 72 non-linear frequency response curves taken at different V_{dc} and V_{ac} voltages (see Supplementary Information Section VII). For each dc voltage we fit 8 frequency response curves at once, and the results are shown in a histogram added in Figure 6b. The error in determining the Young's modulus using the proposed method ranged from 4 – 8 %. The reviewer is correct that further reduction in the error of the mean is possible by making more measurements, as it scales approximately proportionally to $1/\sqrt{N}$, where N is the number of measurements.

To get information on the device-to-device variability, measurements on 2 additional devices have been included. The device-to-device variation, shown in Table 2, is in the order of the standard deviation based on the repeatability measurement. We note that device-to-device variations will be mainly determined by the variation in material and device properties of the nanodrums, and are not indicative of the precision of the characterisation method.

Second, it is a fixed-fixed beam. Why can we not use its linear region of operation with small excitation to find Young's modulus as is the case with most other resonators. The nonlinearity adds interesting aspects and is worth studying; however, its variability and difficulties of stability make it even more difficult to apply in real practice.

The resonance frequency of beams (or circular plates) is indeed related to the Young's modulus of the material, even in the linear regime of operation. For very thin membranes ($h/R < 0.001$), however, the resonance frequency is completely determined by their pre-tension (please see, e.g. ref. 19 equations (1) and (2)), and is independent of the Young's modulus of the material. We therefore need access to the nonlinear regime of operation, where the Young's modulus influences the magnitude of the geometric nonlinearity of the system.

Reviewer #2 (Remarks to the Author):

This is an important paper that describes the use of resonance excitation and laser interferometry to evaluate the Young's modulus of 2D materials. The nonlinear response of the material and a single-parameter fit to the Duffing equation are used. As the authors point out, even though the AFM is most often used, there are outstanding issues with the use of the AFM to extract the Young's modulus of 2D materials. The method proposed in this paper provides an attractive alternative. I suggest the paper be accepted for publication after the authors have addressed the following.

In the Theory section the authors say bending rigidity is neglected, which is a good approximation for $h/R > 1/1000$. However, in the Discussion, they say this approximation is valid for membranes under tension (thickness-to-radius ratio $h/R < 0.001$). The latter is correct.

We thank the reviewer for his/her positive comments and for pointing out this error. It has now been corrected.

Reviewer #3 (Remarks to the Author):

The authors report on a new method to determine the Young's modulus of thin membranes based on graphene and MoS₂. The method consists in measuring the nonlinear dynamic response of the resonator. However, it is not clear whether this method is more reliable than previous methods, since the Young's modulus obtained with their new method is similar to that obtained with the previously existing AFM method (Figs. 2 and 4).

We thank the reviewer for the comments. To be clear, we note that we do not claim that the method is more reliable than previous methods. It is an alternative method that acquires the material properties in a different way. The advantages of the proposed method are that it is a non-contact method and that it operates at higher frequencies. The fact that the obtained Young's modulus is similar to AFM, confirms the reliability of both AFM and the new nonlinear dynamic method.

Moreover, the authors write in the introduction: "values [of the Young's modulus] reported in literature show large spreads [14], which may result [...] from imperfections of the measurement method itself"; but the authors fail to show measurements on multiple graphene and MoS₂ resonators in order to show that they obtain the same Young's modulus.

We agree with the reviewer that our work does not prove that the spread is due to imperfections in the measurement method and for this reason we have removed this statement.

It now reads: "Despite the large number of experimental and theoretical studies [14,15], the exact physics behind the elasticity of 2D materials is still a subject of debate[16]. This debate is mainly motivated by the large spread in values reported in literature ($E = 430 - 1200$ GPa)[14], which has been attributed to variations in the material properties and fabrication techniques[17]. As a consequence, there is a significant interest in methods for characterizing the mechanical properties of 2D materials."

Nevertheless, the reviewer makes a valid point by asking to what extent our method results in the same Young's modulus when it is repeated. This is a point that was also raised by reviewer 1. To characterize the uncertainty of our method we need to decouple it from device-to-device variations. This was done by repeating the same measurement 9 times and calculating the standard deviation. We added the following sentence in the main text:

"By calculating the standard deviation out of nine repeated measurements, the measurement error was determined to be 8 %, which is comparable to other methods for determining the Young's modulus of 2D materials [7]."

Additionally, we added measurements on two more devices: one graphene and one MoS₂ drum. The Young's moduli extracted from these measurements fall within the uncertainty limits of the previous drums (see Table 2).

We also added a new figure (Figure 6) and a section in the Supplementary Information (Section VII) where we show more than 70 additional measurements of the new (graphene) device under varying conditions (dc voltages). We use these data to assess the reproducibility and, at the same time, the robustness of the method. We find that the standard deviation of these measurements is also <8%, which means that our measurement error is not influenced by the measurement conditions.

In addition, the authors write in the introduction "One of the most striking properties of these ultra-thin materials is their high Young's modulus", suggesting that the high Young modulus of thin materials is related to their atomic scale thickness. But this is not correct, since the Young modulus of graphene is about the same as that of graphite.

The reviewer is correct, it is not our intention to suggest that the Young's modulus rises when thinning down the material. Instead we wanted to emphasize that the in-plane Young's modulus of layered materials is relatively high. Although the Young's modulus depends on several parameters, the in-plane arrangement of the atomic bonds in layered materials is one of the potential causes for this. To clarify this, we have changed the sentence into: "One of the most striking properties of these layered materials is their high Young's modulus."

Finally, this method may be interesting as a rapid characterization of the Young's modulus, but this is something interesting for specialists only. For these reasons, I recommend rejection.

The characterisation of the mechanical properties of 2D materials is a topic of wide scientific interest, as can be judged from the fact that reference [7] has been cited for more than 10000 times. Since that time, mechanical characterisation of 2D materials has been performed mostly by AFM (ref. 14). Having an alternative to AFM for mechanical characterisation of 2D materials is thus certainly of interest to a broad community. Moreover, the Young's modulus of 2D materials, especially the one of graphene is still a topic of active theoretical and experimental debate [ref 13, 43-45].

Here are additional comments:

1- The authors start the abstract with "Due to their atomic-scale thickness, the resonances of 2D material membranes show signatures of nonlinearities at amplitudes of only a few nanometers." This is a strange sentence since thicker resonators also feature nonlinear resonances at amplitudes of a few nanometers.

This is correct. We now changed this sentence to: "Due to their atomic-scale thickness, the resonances of 2D material membranes show signatures of nonlinearities at amplitudes of only a few picoNewtons".

2- How reliable is their method in quantifying the Young's modulus? It seems that the value of the Young's modulus depends on the effective mass and the electrostatic force. The authors do not include the mass of the contamination adsorbed on the surface, which is known to considerably contribute to the mass. It is not clear whether a precise determination of the electrostatic force can be made.

The added mass due to contamination is difficult to determine indeed. However, since the AFM measurements are mass-independent (only the thickness h plays a role) and the Young's modulus obtained by AFM matches the one that we measured using our method, this suggests that the added mass in this case is insignificant.

Based on this comment of the reviewer, we added the following text in the discussion:

"In addition, the method requires knowledge of the mass of the resonator, which might be affected by contamination. In the presented data, the close agreement between the extracted Young's modulus and the one determined by AFM (which depends on the thickness and not on the mass of the membrane) suggests that the mass estimate is accurate and effects of contamination are small."

Determining the electrostatic force is mathematically straightforward ($F_{el}(\omega) = \frac{\xi \epsilon R^2 \pi}{g^2} V_{dc} V_{ac} \cos(\omega t)$). However, the calculated force does not always match the force felt by the resonator, because of uncertainties in determining the gap size g (due to membrane slack), the dc voltage V_{dc} (due to residual charge on the 2D flake) and the capacitance of the device (due to fringe fields). In order to cross-check the value of the driving force, we employ a second way of deriving it based on the peak RMS amplitude (x_{RMS}) of the calibrated linear frequency response curves using the following relation: $F_{RMS} = \frac{\omega_0^2 m x_{RMS}}{Q}$. We now included a new section in the Supplementary Information that discusses this procedure in more detail.

3- The authors should include a detailed description of the calibration of the displacement, which is probably not straightforward to do. Again, this will contribute to the uncertainty of the estimation of the Young's modulus.

We thank the reviewer for the good remark. We now included a detailed description of the calibration procedure in the Supplementary Information Section 6. We also discuss the possible errors at high amplitudes stemming from the assumption of a constant transduction factor. At high amplitudes, due to small nonlinearities in the transduction, the uncertainty in the measured displacement due to the calibration is estimated to be less than 0.5%. This is now derived mathematically and discussed in detail in the Supplementary Information.

4- How robust is the determination of the Young's modulus considering that the Q-factor gets reduced at large drives?

The described method is robust, because the whole dataset of non-linear resonance curves can be fit using a single fit parameter k_3 , that is directly related to the Young's modulus, irrespective of the potential amplitude dependence of the Q-factor. This is because we use an equation (eq. 22 of the SI) to determine Q uniquely and independently at each driving power. The observed small amplitude dependence of Q is therefore not expected to affect the robustness. The agreement with AFM is another evidence of this fact.

The determined Young's modulus also does not change significantly when the measurement conditions are changed. The previously discussed statistics on the new sample are a confirmation of this. This is further supported by the quality of the fits of more than 70 nonlinear frequency response curves from the new measurements, which have been now also added in the last section of the Supplementary Information.

Reviewers' comments:

Reviewer #1 (Remarks to the Author):

The authors have undertaken new measurements to study the repeatability of the process, which is very commendable. This also presents the potential error boundaries of their technique. It would be good to comment in numerical and experimental errors in their technique.

However, what I was hoping to see was the spread in extracted Young's modulus over large number of devices. The authors have used data from 2 devices to measure device-to-device variability. That is not a good approach, as one needs to measure from large number of devices to get a feeling of process boundaries, device variability as well as tolerances. 2 devices do not provide a good picture of the statistics.

In their comments, the authors have noted that "device-to-device variations will be mainly determined by the variation in material and device properties of the nanodrums, and are not indicative of the precision of the characterisation method." One hopes this is the case; however, this does not mean that the method over the range of tolerances which one may see in actual devices.

Unfortunately, it is the variation in the devices, which is of interest in any engineering application using Graphene. I was hoping that the authors could measure from a large number of devices characterising the inherent variability on the nonlinearity as well as the Young's modulus. This would have been a very important contribution to the body on knowledge.

Reviewer #2 (Remarks to the Author):

I am satisfied that the authors have adequately addressed the concerns of the reviewers.

Reviewer #3 (Remarks to the Author):

The authors did a great work in their response to my comments/questions. All the points were answered in a very convincing way. I recommend publication.

The authors may consider the following minor comment. In the section VI of the supplementary information, it may be nice to plot the variance of the displacement of thermal vibrations as a function of the laser power in order to show that the laser does not heat the resonator and that the temperature is indeed 293 K. Alternatively, the authors may indicate the power of the laser.

Reviewer #1 (Remarks to the Author):

The authors have undertaken new measurements to study the repeatability of the process, which is very commendable. This also presents the potential error boundaries of their technique. It would be good to comment in numerical and experimental errors in their technique.

We thank the reviewer for the positive feedback. It is indeed important to distinguish between numerical and experimental errors in our method. The numerical error is typically very small (< 0.5 %); the error in our measurements is therefore mainly experimental. We now added a sentence in the main text that clarifies this:

“The numerical error in the Young’s modulus from the fitting procedure is typically <0.5% (defined as the 95 % confidence interval of the fit), as determined from the raw data and the fits, like those shown in the Supplementary Figs. 7 - 15. Our measurement error is therefore mainly experimental.”

However, what I was hoping to see was the spread in extracted Young's modulus over large number of devices. The authors have used data from 2 devices to measure device-to-device variability. That is not a good approach, as one needs to measure from large number of devices to get a feeling of process boundaries, device variability as well as tolerances. 2 devices do not provide a good picture of the statistics. In their comments, the authors have noted that "device-to-device variations will be mainly determined by the variation in material and device properties of the nanodrums, and are not indicative of the precision of the characterisation method." One hopes this is the case; however, this does not mean that the method over the range of tolerances which one may see in actual devices.

We agree with the reviewer that two devices do not provide enough statistics and it was not our intention to claim this. We therefore removed the statement: “...suggesting that the device-to-device variation of the Young's modulus is of the order of the measurement error, similar to what was reported in[7].”

Unfortunately, it is the variation in the devices, which is of interest in any engineering application using Graphene. I was hoping that the authors could measure from a large number of devices characterising the inherent variability on the nonlinearity as well as the Young’s modulus. This would have been a very important contribution to the body on knowledge.

We agree with the reviewer that characterizing the device-to-device variability would be of interest for engineering applications. However, due to the manual transfer process, sample fabrication requires a lot of time and studying enough devices for meaningful statistics (e.g. more than 30 drums) would require many months of work and significant delays. Moreover, we believe that it is beyond the scope of this work, that aims to demonstrate a new characterization method.

Nevertheless, to still convey this interesting suggestion of the reviewer we have added the following sentence to the manuscript:

“The presented method can prove to be useful for fast statistical analysis of the spread in material properties [13,42-44] and variability of device properties in future 2D material-based products.”

Reviewer #2 (Remarks to the Author):

I am satisfied that the authors have adequately addressed the concerns of the reviewers.

We thank the reviewer for the time and effort to read our revised manuscript and for the positive comment.

Reviewer #3 (Remarks to the Author):

The authors did a great work in their response to my comments/questions. All the points were answered in a very convincing way. I recommend publication.

We thank the reviewer for the positive comments and for the time and effort to review our revised manuscript.

The authors may consider the following minor comment. In the section VI of the supplementary information, it may be nice to plot the variance of the displacement of thermal vibrations as a function of the laser power in order to show that the laser does not heat the resonator and that the temperature is indeed 293 K. Alternatively, the authors may indicate the power of the laser.

The reviewer makes a valid point by emphasizing this effect. Choosing the right laser power is always a trade-off between good signal and low heating/photothermal effects. In our measurements we use an incident laser power of 0.42 mW. We now added this value in the Methods section.

Following the Reviewer's suggestion we also added a laser power dependence measurement of the strain in the membrane. Using this data, we were able to extract the mechanical pre-strain and estimate the temperature increase from the added thermal strain (see Supplementary Figure 6 and Supplementary Note 6). We extracted a value of ≈ 0.8 K for the laser-induced temperature increase, which translates to about 0.1 % error in the amplitude calibration.